# Overview of Diagnostic Methods, Disease Prevalence and Transmission of Mpox (Formerly Monkeypox) in Humans and Animal Reservoirs

**DOI:** 10.3390/microorganisms11051186

**Published:** 2023-04-30

**Authors:** Ravendra P. Chauhan, Ronen Fogel, Janice Limson

**Affiliations:** Biotechnology Innovation Centre, Rhodes University, Makhanda 6139, Eastern Cape, South Africa; r.fogel@ru.ac.za

**Keywords:** Mpox diagnostics, Mpox genome sequencing, Mpox host range, Mpox transmission, Mpox, orthopoxvirus, zoonosis

## Abstract

Mpox—formerly monkeypox—is a re-emerging zoonotic virus disease, with large numbers of human cases reported during multi-country outbreaks in 2022. The close similarities in clinical symptoms that Mpox shares with many orthopoxvirus (OPXV) diseases make its diagnosis challenging, requiring laboratory testing for confirmation. This review focuses on the diagnostic methods used for Mpox detection in naturally infected humans and animal reservoirs, disease prevalence and transmission, clinical symptoms and signs, and currently known host ranges. Using specific search terms, up to 2 September 2022, we identified 104 relevant original research articles and case reports from NCBI-PubMed and Google Scholar databases for inclusion in the study. Our analyses observed that molecular identification techniques are overwhelmingly being used in current diagnoses, especially real-time PCR (3982/7059 cases; *n* = 41 studies) and conventional PCR (430/1830 cases; *n* = 30 studies) approaches being most-frequently-used to diagnose Mpox cases in humans. Additionally, detection of Mpox genomes, using qPCR and/or conventional PCR coupled to genome sequencing methods, offered both reliable detection and epidemiological analyses of evolving Mpox strains; identified the emergence and transmission of a novel clade ‘hMPXV-1A’ lineage B.1 during 2022 outbreaks globally. While a few current serologic assays, such as ELISA, reported on the detection of OPXV- and Mpox-specific IgG (891/2801 cases; *n* = 17 studies) and IgM antibodies (241/2688 cases; *n* = 11 studies), hemagglutination inhibition (HI) detected Mpox antibodies in human samples (88/430 cases; *n* = 6 studies), most other serologic and immunographic assays used were OPXV-specific. Interestingly, virus isolation (228/1259 cases; *n* = 24 studies), electron microscopy (216/1226 cases; *n* = 18 studies), and immunohistochemistry (28/40; *n* = 7 studies) remain useful methods of Mpox detection in humans in select instances using clinical and tissue samples. In animals, OPXV- and Mpox-DNA and antibodies were detected in various species of nonhuman primates, rodents, shrews, opossums, a dog, and a pig. With evolving transmission dynamics of Mpox, information on reliable and rapid detection methods and clinical symptoms of disease is critical for disease management.

## 1. Introduction

Mpox, previously known as monkeypox, is a re-emerging zoonotic virus pathogen belonging to the larger orthopoxvirus (OPXV) genus in the *Poxviridae* family [1,2]. Some of the most prominent symptoms of Mpox infections in humans include eruptive skin lesions [3,4,5,6,7,8,9], rash [3,5,6,7,10,11,12,13,14], fever [3,4,5,6,8,11,12,13,14], and lymphadenopathy [5,6,8,10,11,12,14]. These symptoms are largely similar to infections caused by other members of the OPXV genus, such as variola virus (VARV, the causative agent for smallpox disease) [15,16,17,18,19] and cowpox virus (CPXV, causing cowpox disease) [20,21,22,23,24], along with viral infections from other virus families, e.g., chickenpox caused by the varicella-zoster virus (VZV), a member of the *Herpesvirus* family [25,26,27,28,29]. The similarities in clinical manifestations of these diseases in humans make the diagnosis of Mpox disease by observing clinical disease symptoms alone, challenging. Therefore, the current World Health Organization (WHO) guidelines require the confirmation of Mpox diagnosis via reliable laboratory testing to identify infected individuals in order to manage its spread [30,31,32].

Members of the OPXV genus are known to infect a broad range of mammalian hosts [33,34,35,36]; however, the mechanisms behind their ability to transmit between host species are not well understood. Mpox is a highly-pathogenic poxvirus, carrying a human case fatality rate ranging between 1 and 11% [2,37,38,39]. Mpox infections may also result in numerous complications, such as miscarriage and intrauterine fetal death in pregnant women [40,41], vision loss due to corneal lesions [42,43], and encephalitis [44,45], among others. The 2022 multi-country outbreaks of Mpox reported an unprecedented high human-to-human transmission. The occurrence of these outbreaks outside the endemic regions in West and Central Africa [46,47] suggests that Mpox has further adapted to infect humans. Together with the associated case fatality rates and complications, this virus is a re-emerging disease of significant concern [48,49,50].

While the active infection of Mpox was suggested to be first identified in 1958 in cynomolgus monkeys (*Macaca fascicularis*) that were placed in captivity at the Statens Serum Institut in Copenhagen, Denmark, having been received from Singapore [51], a recent study using historical skin samples of African rope squirrel (*Funisciurus* sp.) stored in museums originally collected across Central Africa over the past 120 years provided retrospective molecular evidence of the Central African (formerly Congo Basin) clade of Mpox in specimens from the Democratic Republic of the Congo (DRC) and Central African Republic (CAR) from as early as 1899 [52]. Similarly, molecular evolutionary analysis also indicates the historic presence of Mpox: one study suggests that Mpox may have separated from Old World OPXV approximately 3500 years ago [53], while the separation of the West African Mpox subtype from other clades is estimated to have occurred between 600 years ago [53] and 1200 years ago [54]. These collectively suggest that Mpox disease may have been in circulation much earlier than its first identification.

The clinical features of the Mpox disease in cynomolgus monkeys in 1958 exhibited skin lesions typical to the smallpox virus disease. The investigation using serologic assays determined the presence of OPXV-specific antibodies in sera samples. Electron microscopy (EM) investigation of isolates from these samples observed brick-shaped particles ranging between 100 and 150 nm in size, distinguishable from the other known members of the OPXV genus [51]. Due to its first detection in monkeys, the term ‘monkeypox virus’ was coined in 1958 and, in November 2022, was redesignated “Mpox” by the WHO [55,56].

The first documented case of Mpox infection in humans was in a nine-month-old male in the Democratic Republic of Congo (DRC), formerly Zaire; the patient, presenting fever and a rash, was admitted to a hospital on 1 September 1970 [57]. Similar to its first identification, the clinical manifestation of the disease appeared similar to smallpox—another OPXV-derived illness—which was previously in active circulation in the region [57,58]. Virus isolation from this patient successfully identified the pathogenic agent to be Mpox [58]. Subsequently, between September 1970 and May 1971, six more human cases initially suspected to be smallpox were reported across West Africa: in Liberia, Sierra Leone, and Nigeria. From these, four Mpox virus isolates were recovered and characterized using the reference strains of vaccinia virus (VACV), VARV, and Mpox [59]. From this study, the first distinguishable characteristics in the infection profile between Mpox and other members of OPXV were offered [59]. These distinctions were subtle: along with greater virulence in infected mice and chicken embryos, rabbits intradermally inoculated with isolated Mpox produced more severe illnesses compared to VARV and VACV [59]. Subsequently, various serologic investigations focused on the diagnosis of OPXV infection through the detection of OPXV-specific antibodies produced by infected humans and animals; some of these studies also combined virus isolation and EM study of clinical samples to arrive at a conclusive diagnosis of Mpox disease in humans [60,61,62].

Historically, Mpox was endemic to some countries in West and Central Africa but has recently spread further [63]. The first human cases of Mpox disease outside the endemic regions were reported in 2003, presumably caused by a shipment of small rodents from Ghana to the Midwestern United States as part of the exotic pet trade. These were assumed to transmit Mpox to prairie dogs housed at a pet store in the United States through contact with imported rodents. The infected prairie dogs were purchased by several households in the Midwestern United States [64], resulting in zoonotic transmission of Mpox from the pet prairie dogs to the humans due to close contact within the households [65,66]. A total of 37 laboratory-confirmed cases of Mpox out of 71 suspected were reported in the United States between 15 May and 20 June 2003 [67]. More recently, the exportation of human cases of Mpox from Nigeria to other countries via air travelers was first reported in 2017 [68]. The frequency of imported Mpox cases outside Africa quickly increased in subsequent years [7,10,69,70,71,72], most likely due to the adaptation of Mpox to human hosts [73]. By April–May 2022, multi-country outbreaks of Mpox erupted, which witnessed an unprecedented increase in human infections, with many cases of human-to-human transmission reported due to sexual contact [48,74,75,76,77,78,79]. According to the WHO, between 1 January 2022 and 27 February 2023, a total of 86,173 laboratory-confirmed cases of Mpox, including 99 deaths, were reported from 110 countries [80].

Rapid, reliable, and scalable testing methods for Mpox detection are imperative for mitigating multi-country Mpox disease outbreaks by minimizing human-to-human transmission. Currently, Mpox disease diagnosis constitutes a combination of clinical examination and molecular detection with real-time PCR or conventional PCR followed by partial genome sequencing using clinical samples, such as surface lesion swabs or crusts [81]. This review discusses the technological advancements in Mpox detection in naturally infected human and animal populations, along with clinical manifestations and transmission dynamics of the disease.

## 2. Methodology

### 2.1. Article Search Criteria

The guidelines of Preferred Reporting Items for Systematic Reviews and Meta-Analyses (PRISMA) [82] were followed to identify the relevant records for inclusion in the study (Figure 1). The original research articles and case reports documenting natural infections of Mpox in humans and animals were identified using Google Scholar and NCBI-PubMed databases. Search terms (Table 1) were entered one by one in the databases, and the titles of the suggested articles were screened to identify the relevant reports published up to 2 September 2022. To enhance record retrieval, search terms were not enclosed in quotation marks. The relevant articles thus identified were downloaded on a computer drive for further screening for inclusion in the study.

### 2.2. Article Inclusion and Exclusion Criteria

The original research articles and case reports that documented the natural occurrence of Mpox disease in humans and animals up to 2 September 2022 were included in the study. Review articles, experimental studies, articles in a language other than English, abstracts, and conference proceedings were not included in the study.

## 3. Results

Search terms used for record retrieval (Table 1) were not enclosed within quotation marks; this was intended to increase the variations of the terms used by the search engines when retrieving records. In total, 1397 records were screened in NCBI-PubMed and 5527 records in Google Scholar. These two databases routinely index the cases and reports of Mpox disease globally, including the weekly Mpox disease updates from the WHO. Titles and abstracts were manually read to evaluate the relevance for the further full-text reading, focusing strictly on case studies and original research articles that applied diagnostic technologies to the detection of Mpox were included: of the 6924 studies initially screened, only 104 relevant records were found eligible and thus included in this study (Figure 1).

### 3.1. Trends of Mpox Disease Investigations

A total of 104 full-text research articles and case reports published up to 2 September 2022 documenting natural infections of Mpox in humans and animals were retrieved (Figure 1). The chronological profile of these included studies is presented in Figure 2, where it is compared to both Mpox case numbers which are included in this study, and to SCOPUS-retrieved records for the term (“monkeypox” AND “human”, followed by either “detection”, “characterisation” or “diagnosis”), as a general measure of research conducted in this field. Figure 2 shows that, in general, the included studies used further in this paper corresponded well to the overall literature being published surrounding this topic (SCOPUS records); in general, a larger number of included articles were retrieved when research around this topic increased. This indicates that the included studies are representative of the evolving manner by which Mpox was, and currently is, being monitored. A large subset of total records retrieved were either research reviews on existing literature or the development of novel detection methods and experimental studies that were never applied to real-world samples of Mpox nor used in the diagnosis of infection. Therefore, the present study included exclusively the original research articles and case reports that detected natural infections of Mpox in human and/or animal populations. The aim of the study was to compare various diagnostic technologies used for Mpox detection in clinical samples, analysis of evolving clinical symptoms and signs of disease in hosts, and transmission dynamics.

Comparing research interest (via SCOPUS-retrieved records) with case incidences (Figure 2) shows that a sharp rise in Mpox investigations occurred during the last two decades after Mpox outbreaks in the Midwestern United States in 2003 [44,64,65,66,89,90,91,92], i.e., the reporting of human cases of Mpox in a non-endemic region. This is again evident in the spike in published literature shortly after 2017, when several reports of Mpox in air travelers appeared from many non-endemic countries [4,68,69,71,93] and more recently during the multi-country outbreaks of Mpox, starting in April–May 2022 [14,94,95,96,97,98,99,100]. These are linked to the reportedly unprecedentedly high rates of human-to-human transmission, in turn causing the highest number of Mpox human cases.

Based on the Mpox genome sequences and clinical features of the disease, two distinguishable clades of Mpox were identified circulating before the 2022 Mpox outbreaks. These were the West African and Central African (formerly, Congo Basin) clades, based on their geographic origins [101,102]. The phylogenomics analysis of available Mpox genomes including the genomes of the 2022 Mpox outbreaks suggested that the 2022 Mpox outbreaks were caused by a newly evolved clade termed ‘hMPXV-1A’ lineage B.1 [103]. Historically, more frequent circulation of the West African clade than the Central African clade was observed before the emergence of the 2022 Mpox outbreaks (Figure 3).

### 3.2. Methods Used for Mpox Detection in Clinical and Tissue Samples of Humans and Animals over the Period and Their Significance

Technological advancement over the decades since Mpox was first characterized has strongly influenced the range of available methods for the detection of Mpox. Mpox investigations have used a variety of methods for the confirmation of disease (Figure 4); most studies focused on a single—or a limited selection of separate methods—to process samples, but a few studies used a combination of various techniques. These have been separately categorized into approaches that diagnose Mpox in samples via the detection of virus genomes (overwhelmingly using PCR-based approaches), serological methods aimed at detecting virus-specific antibodies, such as Enzyme-Linked Immunosorbent Assay (ELISA) and hemagglutination inhibition (HI) assays, and more historic approaches that are still being applied today, such as immunohistochemistry (IHC), virus isolation, and electron microscopic identification of virus particles.

The current WHO guidelines require molecular detection of Mpox using Mpox-specific real-time PCR or conventional PCR in combination with genome sequencing for epidemiological analysis [32]. The detection of viral genomes in samples using oligonucleotide primers and a probe for real-time PCR offering a rapid, sensitive, and specific detection of the target has become the most popular method for Mpox disease confirmation using clinical samples over the past 20 years (*n* = 48 studies; Figure 4), while conventional PCR-based approaches enjoyed similar popularity (*n* = 32 studies). Several studies combined molecular detection using OPXV or Mpox-specific real-time PCR/conventional PCR-genome sequencing to either distinguish Mpox from other OPXVs or to identify specific clades of Mpox. PCR amplification and Sanger sequencing of the amplicon [9,89,109] were used for characterization and to distinguish Mpox from other OPXV using stretches of its hemagglutinin (HA) gene [65,90,107,115,116]. Next-generation sequencing technologies, such as Ion Torrent PGM [106,116], Illumina [4,9,68,104,107,112], and Oxford Nanopore MinION [4,68,117,118,119], are cutting-edge approaches being employed for Mpox diagnosis (*n* = 16 studies). Next-generation sequencing, in particular, offers the opportunity for whole genome sequencing of Mpox in clinical samples, in turn, providing insight into the phylogenomics of circulating Mpox strains. Next-generation sequencing-based studies conducted during 2022 Mpox outbreaks identified the emergence of a new Mpox clade, termed ‘hMPXV-1A’ lineage B.1 [103]. The advances in the cost-effectiveness of next-generation sequencing are expected to further improve its routine application, in turn driving the molecular characterization of Mpox in patients and providing enhanced epidemiological and evolutionary data for analyses.

In addition to molecular approaches, many studies have used serological approaches to identify Mpox. These largely monitor the prevalence of the virus in patients via the production of virus-specific antibodies during and post-infection, using secondary antibody-based systems to detect virus-specific antibodies. Diagnosis using either one or a combination of ELISA-based monitoring of the production of Mpox-specific IgG (*n* = 23 studies) [3,11,44,92,120,121,122,123,124] or IgM antibodies (*n* = 11 studies) [11,44,83,92,120,121,123,124], neutralization assay (*n* = 5 studies) [113,125,126], HI (*n* = 8 studies) [60,61,62,113,125], and/or complement fixation assays (*n* = 2 studies) [60] in blood samples have been reported.

In general, molecular detection methods have several advantages over serological methods. Primarily, molecular detections are routinely capable of differentiating between Mpox and other species of OPXV in a multiplex assay [127,128], unlike most serological assays used, which due to immunological cross-reactivity between human OPXV, offer limited diagnostic value [129,130,131]. Although a few studies used Mpox-specific antibodies, they may lack reproducibility and are complex to execute [132,133,134]. Additionally, most serologic assays tend to offer only limited usage for the detection of acute infection due to the persistence of circulating antibodies post-infection [135], which may result in misleading diagnostic outcomes. The use of multiple sample types from a patient is suggested for IgG-based serologic investigations when determining recent exposure, comparing the levels of antibodies in those samples to achieve diagnosis [124]. IgM-detecting ELISA offers detection of recent OPXV, as these types of antibodies are markers of acute immune responses [124]. While IgG ELISA is useful in the detection of vaccine efficiency testing or correlating epidemiologically defined outbreaks, it tends to be less efficient in the context of species differentiation of OPXV [136]. An advantage that serological tests tend to offer over molecular detection is the longer timeframe in which exposure to Mpox can be detected. While PCR-based detection methods require the availability of sufficient virus genome copies (mainly confined to rash, fluid, or crust specimens), serologic detection, using blood, offers flexibility in terms of sample collection and can detect prior exposure after the clinical symptoms of disease subside, allowing for retrospective diagnosis.

The final category (virus isolation and EM) has been consistently applied throughout the known existence of Mpox (Figure 4). Virus isolation, coupled with EM-based morphological identification of virus particles, was instrumental in the initial characterization of the virus and its first report of human infection [58]. While these are not the recommended test methods for routine diagnosis of Mpox in clinical laboratories [32], they have still been employed in recent studies (Figure 4). Virus isolation remains useful in virological investigations that report on the environmental persistence of the virus [137], identification and characterization of poxviruses [58,138], or investigation of the pathophysiology and virulence of viruses, including Mpox, under experimental settings [139].

Similarly, EM offers a great advantage for detecting emerging or novel viruses in clinical samples, as it does not rely on virus-specific reagents and can provide unbiased detection of viruses in general within a sample. For this reason, EM remains a method of choice for determining the size and shapes of poxvirus particles and distinguishing among poxviruses morphologically. Several studies during 2022 Mpox outbreaks used EM [14,96,140] and virus isolation [98,99,138,140] to confirm diagnoses, especially for patients where transmission may have occurred due to sexual contact; these patients presented with clinical disease symptoms similar to those of sexually-transmitted diseases. It can be observed from the studies that EM [8,105,110,140,141,142] and virus isolation [8,12,13,58,95,99,105,113,120,140,141,142] have been consistently used over the decades within and outside the endemic regions for the confirmation of Mpox diagnosis.

Other methods making use of antibodies to detect portions of virus particles (Immunographic methods, Figure 4) such as immunofluorescence (*n* = 8 studies) [125,126,130], immunohistochemistry (*n* = 7 studies) [40,44,64,92,97], radioimmunoassay (*n* = 5 studies) [60,62], agar-gel test (*n* = 3 studies) [59,61], precipitation-in-gel test (*n* = 3 studies) [60], and Western blot (*n* = 3 studies) [113,143] were used historically, but have become less prevalent in recent years, except for immunohistochemistry (IHC) which offers tissue-based diagnostics with specificity, sensitivity, and great reproducibility [144]. The IHC remains a preferred method for investigating systemic Mpox infections in various human tissue samples in real-time, as well as retrospectively. Many studies in the recent past, including during the 2022 Mpox outbreaks, successfully detected Mpox antigen in clinical samples using IHC [11,44,64,90,97].

Briefly, our analyses indicated that molecular detection of Mpox using PCR or real-time PCR in combination with Sanger and next-generation sequencing offers rapid, sensitive, and reliable detection of Mpox in human clinical samples. In addition, standard virological methods, such as virus isolation and EM, as well as tissue-based detection, such as IHC, are still commonly used in research laboratories for the detection and confirmation of Mpox in human clinical samples or tissues. The IgG and IgM ELISA remain the most used serologic methods for Mpox or OPXV antibody detection in human sera.

### 3.3. Comparison of Detection Methods by Prevalence of Use in Human Mpox Diagnosis

The majority of the selected studies (*n* = 104) investigated natural infections of Mpox in humans (*n* = 83 studies). The numbers of human-derived samples processed were extracted from all included studies in this review and categorized by the detection approaches outlined in Section 3.2 above. Figure 5 summarises the aggregated number of samples processed using the different methods, as well as the number of positive diagnoses made with each method. While real-time PCR detected most Mpox-positive samples, conventional PCR, virus isolation, EM, IgG and IgM ELISA, HI, and neutralization assays were other, more commonly used, methods than other serologic and immunographic methods.

Overwhelmingly, most samples processed for Mpox detection between 2010 and 2022 used either molecular or genomic detection of viral nucleic acid or antibodies using serological methods. Together, these two approaches comprise 15,140/18,060 (83.8%) patient samples processed in the selected articles and corresponded to 5720/6337 (90.2%) of all positive diagnoses made. Nearly half of the Mpox-suspected clinical samples were processed using either PCR- or real-time PCR-based methods, correlating to the larger number of recent reports that used these techniques (Figure 4) due to these methods’ rapidity, sensitivity, scalability, and efficiency for selectivity to Mpox (discussed in more detail in Section 3.7, below). The overall positivity rate of Mpox in tested human populations was much higher using real-time PCR-based detection (3982/7059; 56.41%) than conventional PCR-based detection (430/1830; 23.49%).

For similar reasons of the rapidity of response and scalability, serological methods comprise the largest remaining group of samples processed. Many studies used only serologic detection of Mpox; most of these studies were conducted before the advent of PCR-based detection. Mpox antibodies were detected in human sera using IgG ELISA (652/1285; 50.74%), IgM ELISA (52/1077; 4.83%), neutralization assay (44/206; 21.36%), HI assay (15/195; 7.69%), and immunofluorescence assay (IFA) (15/164; 9.15%). Detection of anti-Mpox IgG via ELISA showed a higher apparent sensitivity (891/2801; 31.81%) compared to the other serological methods: IgM ELISA (241/2688; 8.96%), HI assay (88/430; 20.46%), and neutralization assay (85/285; 29.82%). The apparent higher sensitivity of IgG ELISA can be attributed to the presence of IgG antibodies post-infection for a longer period than IgM antibodies and can be detected retrospectively. Virus isolation (228/1259; 18.10%) and EM (216/1226; 17.61%) were more commonly used for Mpox or OPXV detection in human clinical samples compared to immunographic methods (Figure 5).

Since most studies used more than one method for Mpox detection, their positivity rates for such combinations of methods are discussed in detail in the following section. Detailed information on Mpox- or OPXV-positive human samples from individual articles selected in this study can be seen in Appendix A. Briefly, studies that attempted only molecular detection of Mpox or OPXV resulted in a higher positivity rate for real-time PCR-based detection (3722/5893; 63.16%) than the conventional PCR-based detection (365/1167; 31.28%). Interestingly, conventional PCR-based methods are frequently included in approaches that use a combination of methods to detect Mpox or OPXV, such as genome sequencing. In a subset of studies, real-time PCR (9/9; 100%) and IHC (11/11; 100%) had higher positivity rates for Mpox detection in a limited sample size. In other studies, conventional PCR (10/14; 71.42%), EM (20/29; 68.96%), and virus isolation (10/22; 45.45%) resulted in a relatively lower positivity rate for Mpox detection in tested samples.

On the other hand, few studies used a combination of serologic and molecular tools for Mpox detection in human samples. While real-time PCR yielded a higher positivity rate for Mpox detection in these samples (126/660; 19.09%) than conventional PCR (16/369; 4.34%), the IgG ELISA (75/736; 10.19%) was more frequently used serologic method for Mpox detection than IgM ELISA (3/8; 37.5%) and IFA (7/7; 100%).

Many studies attempted virus isolation in combination with serologic and molecular investigations for Mpox detection. Such studies, collectively, reported Mpox detection using real-time PCR (122/494; 24.70%), conventional PCR (17/255; 6.67%), Western blot (65/90; 72.22%), IgG ELISA (97/195; 49.74%), IgM ELISA (27/458; 5.89%), neutralization assay (41/79; 51.90%), HI assay (54/77; 70.13%), radioimmunoassay (RIA) (5/23; 21.74%), and IFA (7/7; 100%).

A few other studies, in combination with serologic and molecular methods, used virus isolation, IHC, and EM for Mpox detection. These studies reported the prevalence of Mpox in human samples using IgM ELISA (159/1145; 13.89%), IgG ELISA (67/585; 11.45%), virus isolation (201/1203; 16.71%), EM (196/1197; 16.37%), HI assay (19/158; 12.02%), precipitation-in-gel test (20/47; 42.55%), RIA (17/47; 36.17%), complement fixation test (3/47; 6.38%), IHC (17/29; 58.62%), conventional PCR (22/25; 88.0%), real-time PCR (3/3; 100%), agar-gel test (8/9; 88.89%), and IFA (1/1; 100%). Detailed information on various aspects of Mpox disease, such as clinical disease symptoms, transmission status, and history of contact with animals, along with the number and types of samples and methods used for Mpox detection in naturally infected humans, is provided in Appendix A.

Most serologic assays used ELISA for IgM and IgG detection in human sera. Interestingly, Mpox- specific IgM antibodies in human sera could be detected as early as 5 days after the onset of symptomatic rashes and as long as 56 days after symptom onset [12]; Mpox IgM titer was also detectable for the same duration [123]. In many studies, patients with elevated anti-OPXV IgM titer in blood collected within 56 days after the onset of rash illness were considered confirmed Mpox-positive [11,13]; however, a few sera samples had detectable anti-IgM titers up to 70 days post-onset of clinical symptoms [13]. The IgM ELISA could not detect anti-IgM titer in sera samples collected on day 1 and day 147 after the onset of the rash illness. Of note, sera obtained between 5 and 77 days after the onset of rash illness resulted in 92% sensitivity and 100% specificity for IgM detection [124]. It was intriguing to note that IgM response was suggested to be detectable up to six months in some of the primary vaccinees, based on anecdotal experiences with the Centers for Disease Control and Prevention (CDC) vaccines, based on unpublished data mentioned in one study [5].

Serologically, in human cases where anti-OPXV IgG ELISA was positive and anti-OPXV IgM ELISA was negative, when a blood sample was collected at least 56 days after the onset of rash illness, given that the patient was not previously vaccinated for smallpox, was considered confirmed Mpox-positive [121]. While IgG ELISA failed to detect anti-IgG titer in sera samples that were collected within 3 days and on the 8th day after the onset of rash illness [124], anti-IgG titer in blood at day 12 after the onset of rash illness was detected to be 1:800, which increased four-fold to 1:3200 on day 55 of rash onset [109].

Karem et al. 2005 [124] suggested that the sera samples collected at ≥ 5 days for IgM detection after the onset of rash illness were most efficient for Mpox detection. Of note, the sera samples collected less than 14 days after the onset of rash illness may be problematic for IgG detection in unvaccinated individuals. Of note, sera samples collected after 14 days of rash onset resulted in 100% sensitivity and 88.5% specificity for IgG detection [124].

On the other hand, OPXV- and Mpox-specific real-time PCR failed to detect Mpox DNA in blood samples collected between 61 and 90 days after the onset of rash illness [107]. While most PCR-based studies successfully detected Mpox DNA in clinical samples or blood collected up to day 12 after rash onset [109], only one study reported the detection of Mpox DNA in skin lesions, swabs, and blood obtained from one patient, after 41 days of rash onset [107].

While Mpox whole genome sequences could be obtained using clinical samples with Cq values < 30 [96], one study determined that clinical swabs with Cq values ≥ 35 had very low or no infectivity in tissue culture [145]. Samples obtained during the first nine days after the onset of clinical symptoms had higher viral loads, with Cq < 25 [96]. Many studies used Sanger sequencing for Mpox partial genome sequencing, including HA gene, for confirmation and clade identification; various recent studies used next-generation sequencing methods, such as ION Torrent PGM [116], Illumina sequencing [14,68], or Oxford Nanopore MinION [68,117] for Mpox whole genome sequencing, for confirmation and characterization. The virus isolation [68,95,98,105,110,138] and EM [105] remain useful for confirmation of diagnosis, including those cases where Mpox transmission occurred via sexual contact, and based on clinical symptoms, the cases were misdiagnosed [14]. The overwhelming majority of Mpox human cases were symptomatic; only one study reported two patients with no clinical symptoms [95]. Novel and unusual clinical symptoms were observed in many cases during the 2022 Mpox outbreaks [48]—themselves uncharacteristic of historical Mpox—such as severe anal pain [99,146], proctitis [48,147], and genital ulcers [117]. Of note, these patients were largely thought to have contracted Mpox via human-to-human transmission during sexual contact [14,99,146,148].

While Mpox is endemic in the regions of West and Central Africa, the first report of human Mpox outside its endemic regions appeared in the United States in 2003 when a child was detected as Mpox positive [66]. The outbreak in the United States was the zoonotic transmission of Mpox from infected prairie dogs to humans. Of note, the first human-mediated exportation of Mpox from Africa to outside occurred via air travel in 2017, when four people (two Nigerian nationals, one British, and one Israeli citizen) traveled outside Nigeria [68]. A sharp increase in human cases of Mpox was seen outside the endemic regions, starting after 2017, and culminated in an epidemic starting in early 2022.

### 3.4. Prevalence of Mpox Disease in Animals

Active infection of Mpox was first detected, confirmed, and characterized using lesion swabs obtained from sick cynomolgus monkeys, using HI assay, virus isolation, and EM [51]. Of the selected 104 reports, a substantial minority (*n* = 21 studies) [44,51,58,64,89,91,106,107,111,113,120,122,125,126,130,149,150,151,152,153,154] detailed the diagnosis of Mpox in animal hosts. The animal samples were processed using the different approaches detailed above, with some notable biases evident depending on the species of the host. We observed that only a few animal species, such as prairie dogs [44,64,91], a few species of nonhuman primates [51,106,107], a dead squirrel, and a dog [151,154] had visible external clinical signs of disease at the time of investigation. Many animals under investigation, including various species of rodents [111,113,120,122,130,150,153] and some nonhuman primates [125,149,150], did not exhibit clinical signs of disease. This appears to have led to the selection of a significantly smaller number of animal samples that were collected for Mpox or OPXV detection over the period.

Rather than presenting aggregate numbers of Mpox- or OPXV-positive animal samples detected via different techniques, we categorized the animal species under study in separate groups to compare the Mpox or OPXV positivity in various animal hosts. Real-time PCR (1/189), conventional PCR (62/492), virus isolation (36/54), EM (3/3), IgG ELISA (12/81), HI assay (28/206), neutralization assay (5/14), and IFA (8/17) detected Mpox or OPXV infections in various species of nonhuman primates (Figure 6A; Appendix A). The IgG ELISA (65/736), real-time PCR (34/503), and conventional PCR (26/211) were more frequently used methods to detect Mpox infections or antibodies in rodents than other methods, such as IFA (12/160), neutralization assay (13/43), virus isolation (23/31), EM (3/3), and IHC (2/2) (Figure 6B). While IgG ELISA (17/93) and neutralization assay (2/4) detected Mpox antibodies in the sera of several mammals, including shrews and opossums, real-time PCR (3/70) and virus isolation (1/1) detected active Mpox infections only in a few samples tested (Figure 6C). Intriguingly, Mpox DNA was detected, using PCR, in some of the flies and maggots (5/89) [107] that were feeding upon a dead chimpanzee who had typical Mpox lesions externally (Figure 6D). Detailed information on Mpox prevalence, animal species naturally infected, samples used for detection, methods used, and the clinical signs of disease observed in various animal species is provided in Appendix A.

Most studies that investigated Mpox in animals sampled multiple species of nonhuman primates and rodents, which are known to serve the animal reservoirs of Mpox, irrespective of clinical signs of disease. Intriguingly, many species of rodents with no external clinical signs of disease were found positive for Mpox DNA using samples obtained from multiple internal organs, suggesting systemic infections [111,120]. Mpox DNA was also detected in the multiple internal organs (necropsy tissues) of nonhuman primates with clinical signs of disease [106,107]. In addition, anti-OPXV IgG antibodies were detected in the blood samples of rodents [122,130,150] and nonhuman primates with no external lesions [58,125,126,149,150,152]. Detection of Mpox DNA using PCR, occasionally in combination with virus isolation, using multiple organ tissues and clinical samples obtained from nonhuman primates and rodents such as prairie dogs, wild squirrels, dormice, and shrews suggested the evidence of systemic Mpox infections in these animals. This may justify why nonhuman primates and various species of rodents are considered reservoirs for Mpox infections. The next sections will discuss the known host ranges, clinical signs and symptoms of Mpox disease, and the transmission dynamics.

### 3.5. Host Ranges of Mpox

Apart from humans (*Homo sapiens*), Mpox has a well-documented and broad animal host range, including various species of nonhuman primates and other mammals, such as rodents, shrews, and opossums (Appendix A). Studies investigating nonhuman hosts included: humans and prairie dogs (*n* = 1 study) [44]; humans and rodents (*n* = 2) [113,120]; a human and a dog (*n* = 1) [154]; rodents (*n* = 8 studies) [64,89,91,111,122,130,151,153]; nonhuman primates (*n* = 7) [51,58,106,125,126,149,152]; nonhuman primates and rodents (*n* = 1) [150]: nonhuman primates and insects (*n* = 1) [107]. Using various methods, such as molecular, serologic, and immunographic approaches, these studies identified natural infections of Mpox in cynomolgus monkey (*Makaka fascicularis*) [126,152], chimpanzee (*Pan troglodytes verus*) [107], red colobus monkey (*Colobus badius*) [125], sooty mangabey (*Cercocebus atys*) [106], langur monkey (*Semnopithecus* spp.) [152], vervet monkey (*Chlorocebus pygerythrus*), lesser white-nosed monkey (*Cercopithecus petaurista*) [125], Philippine long-tailed macaque (*Macaca philippinensis*) [152], chacma baboon (*Papio ursinus*) [150], and the yellow baboon (*Papio cynocephalus*). Other mammals identified as naturally infected with Mpox include domestic pig (*Sus scrofa domesticus*) [113], dog (*Canis lupus familiaris*) [154], gray short-tailed opossum (*Monodelphis domestica*) [153], southern opossum (*Didelphis marsupialis*) [153], shrew (*Crocidura littoralis*) [150], and elephant shrew (*Petrodromus tetradactylus*) [122]. Natural Mpox infections have been widely reported in several species of rodents, including African hedgehog (*Atelerix* spp.) [153], African giant pouched rat (*Cricetomys* sp.) [150], giant-pouched rat (*Cricetomys emini*) [3], African dormouse (*Graphiurus lorraineus*) [153], target rat (*Stochomys longicaudatus*) [111], rusty-nosed rat (*Oenomys hypoxanthus*), wild mice (*Mus musculus*) [130], wild rats (*Rattus norvegicus* and *Rattus rattus*) [130], gerbil (*Gerbillus* sp.) [89], prairie dog (*Cynomys* sp.) [44,89], black-tailed prairie dog (*Cynomys ludovicianus*) [153], chinchilla (*Chinchilla *sp.) [89], jerboa (*Jaculus* spp.) [153], hamster (*Cricetus* spp.) [89], woodchuck (*Marmota monax*) [153], and several species of squirrels, such as Thomas’s rope squirrel (*Funisciurus anerythrus*) [111], Lunda rope squirrel (*Funisciurus bayonii*) [111], Kuhl’s tree squirrel (*Funisciurus congicus*) [113], and sun squirrel (*Heliosciurus rufobrachium*) [113,122]. Interestingly, Mpox DNA was also detected in maggots and flies that were feeding upon a dead chimpanzee (*Pan troglodytes verus*) in the wild, reported in one study [107].

### 3.6. Clinical Manifestations of Mpox Disease in Humans and Animals

Many studies reported clinical disease symptoms and clinical signs that occurred due to Mpox infections in humans and animals, respectively. Of note, rash (*n* = 64), fever (*n* = 57), skin lesions (*n* = 48), lymphadenopathy (*n* = 38), headache (*n* =23), myalgia (*n* = 17), sore throat (*n* = 15), chills (*n* = 12), malaise (*n* = 10), and fatigue (*n* = 10) were some of the common disease symptoms in humans (Figure 7A; Appendix A). The Mpox disease resulted in clinical signs of anorexia, lethargy, wasting, and influenza-like illness, among other clinical signs in prairie dogs (*Cynomys* sp.) [64,65,91] (Figure 7B; Appendix A). Interestingly, no clinical signs of disease were reported in other Mpox-positive rodent species and shrews. Clinical signs of rash, vesicular lesions, respiratory illness exhibited by coughing, breathing with mouth open, and nasal and ocular discharge were observed in some of the chimpanzees (*Pan troglodytes verus*) [107]. A sooty mangabey (*Cercocebus atys*) that was found dead was Mpox-positive and had skin lesions [106]. Imported cynomolgus monkeys (*Makaka fascicularis*) from Singapore, in captivity at the Statens Serum Institut, Copenhagen, Denmark, where the Mpox disease was first detected in animals in 1958, had maculopapular rash and skin lesions [51] (Figure 7C; Appendix A). A four-year-old greyhound pet dog (*Canis lupus familiaris*) had mucocutaneous lesions, abdomen pustules, and anal ulcers [154]. Skin eruptions were observed in a wild squirrel (*Funisciurus anerythrus*) that was found dead near a village in DRC. Interestingly, Mpox was confirmed with virus isolation; the first declared Mpox isolation from a dead squirrel [151].

### 3.7. Oligonucleotide Primers Used for Mpox Detection in PCR and Real-Time PCR Assays

Some of the commonly reported oligonucleotide primers and probes that were used for the molecular detection of Mpox, targeting specific genes, using real-time PCR or conventional PCR in previous studies, are summarized in Table 2.

While PCR-based detection offers a great advantage over serological assays, often regarding sensitivity and specificity, for the detection and identification of infectious pathogens in test samples, it also has some serious limitations. For example, PCR-based detection of infectious pathogens, including Mpox, requires post-PCR visual quantification with agarose gel electrophoresis. Independently of the initial concentration of the DNA template used for amplification of the target, the conventional PCR assay might fail to amplify enough of the DNA required for visual quantification on the agarose gel, which might result in a false negative [162]. Most conventional three-step PCR assays use an elongation temperature of 72 °C, which is the recommended temperature for optimum activity of Taq polymerase enzyme in a PCR assay. The conventional PCR assay is usually designed to generate relatively large sizes of amplicons, required for post-run quantification on the agarose gel. The higher elongation temperature in PCR should ideally melt any secondary structures that might be formed in the long DNA template formed during amplification and might block the extension of the amplified product [162].

Such limitations have been greatly overcome by the use of probe-based, such as TaqMan, real-time PCR for highly sensitive and specific detection of the target in a test sample. While several TaqMan real-time PCR assays may use a lower extension temperature, such as 60 °C [163], they are also designed to generate much shorter amplicons than the conventional PCR, thus minimizing the probability of forming secondary structures during amplification and extension. Another great advantage that TaqMan real-time PCR detection of Mpox offers over conventional PCR is its higher specificity and sensitivity due to the use of fluorescently labeled probes.

The real-time PCR assays have used various fluorescent probes for Mpox detection, which have covalently bonded fluorescent-labeled reporter dye at the 5-prime end, such as tetrachlorofluorescein (TET) [13,64,114,153,156] or 6-carboxyfluorescein (FAM) [3,8,114,157,158] and a quencher, such as blackhole quencher 1 (BHQ1) [3,8,114,157,158], quencher succinimidyl ester (QSY7) [13,64,114,153,156], or minor groove binding (MGB) [40,89,109,159] at the 3-prime end. In the event of the amplification of the target molecule during the reaction, the given probe anneals to the target, followed by its cleavage by the Taq polymerase enzyme [164]. The cleavage or degradation of the fluorescent probe results in the emission of a fluorescence signal based on the principle of fluorescence resonance energy transfer (FRET) [165]. In the lack of amplification of the target molecule, the fluorescent probe remains intact, and as a result, its proximity with the quencher significantly decreases the emitted fluorescence. Therefore, by monitoring the fluorescence intensity during a reaction, real-time PCR determines the amplification of the target [164]. Interestingly, the MGB probe, due to its chemistry, offers a higher melting temperature and increased specificity to the target compared to other probes used [166].

Often, real-time PCR quantifies the concentration of the target as the cycle threshold (Ct) value by determining the lowest PCR amplification cycle at which significant amounts of fluorescent signals are produced. Higher concentrations of targets thus correspond to lower Ct values and vice versa; in general, lower Ct values correspond to increased infectivity and higher viral load [167]. Paran et al. 2022, reported that Ct values ≥ 35 in clinical samples predict very low or no infectivity of Mpox and correlates with noninfectious virus [145]. Lapa et al. 2022, isolated the replication-competent Mpox from the seminal fluid with a Ct value of 22.7 [138]. In another study, Ma et al. 2022, reported isolation of Mpox from 19 human clinical samples that had Ct values in the range of 15.3 to 29.0 [168].

The selected studies for Mpox detection in humans used a wide range of clinical samples; however, the preferred human samples included skin lesions, lesion swabs or exudates, vesicular lesions, vesicular fluids or swabs, crusts, and pustule swabs for molecular detection and blood or sera for serologic detection. Detailed information on the sample types and Mpox positivity is provided in Appendix A for human samples and Appendix A for animal samples.

Mpox comprises a double-stranded DNA genome of approximately 197 kb [4,169] and encodes 190 open reading frames (ORFs) [169]. Stretches of nucleotide sequences in some of the gene segments of orthopoxviruses, for example, P4A, a major core protein 4a precursor, is a suitable target, offers pan-orthopoxvirus detection, using clinical samples [155]. Another most used target includes conserved sequences of E9L, the DNA polymerase gene, which offers non-variola orthopoxvirus detection in clinical samples with real-time PCR [13,64,114,153,156]. Table 2 summarizes the most used oligonucleotide primers and probes that offer sensitive and specific detection of Mpox in clinical samples. In fact, the confirmatory diagnosis of Mpox requires the detection of Mpox DNA in clinical samples using Mpox-specific real-time PCR or conventional Mpox-specific PCR in combination with partial genome sequencing for clade determination and epidemiological analysis [32]. While most studies adopted a two-step approach to Mpox detection, using OPXV-specific real-time PCR for orthopoxvirus detection followed by Mpox-specific real-time PCR or Mpox-specific conventional PCR with partial genome sequencing for confirmation of Mpox DNA, a study used a novel multiplex real-time PCR assay, included specific-Mpox signature encoded in Mpox envelop gene (B6R) and the OPXV-specific signature encoded in the DNA polymerase gene (E9L, non-variola) [64]. Interestingly, an automated Mpox testing system termed GeneXpert MPX/OPX platform, though in its initial phase of development [128], can detect Mpox DNA using OPXV-generic and Mpox-specific multiplex detection approach with high sensitivity and 100% specificity [128]. The availability of such a high-throughput and automated detection system would certainly facilitate scalable detection of Mpox under field outbreak settings with the added advantage of minimal manual sample handling to reduce the risk of exposure to the operator.

### 3.8. Transmission Dynamics of Mpox Disease

Reports studying the zoonotic and interspecies transmission of Mpox are summarised in Figure 8A. The research articles and case reports investigated in the present study identified 27 studies, mostly before 2020, that documented human-to-human household transmission of Mpox presumed due to close, extended contact between patients [3,5,7,9,12,13,39,42,44,60,61,62,68,71,83,85,87,104,113,121,142,143,158,170,171,172,173]. In contrast to these reports, the 2022 multi-country outbreaks of Mpox were dominated by human-to-human transmission via sexual contact, as reported in 18 studies [14,31,48,86,94,95,96,99,116,138,146,148,174,175,176,177,178,179], raising concerns about Mpox’s potential as a sexually-transmitted infection. Co-infection with sexually-transmitted infections was also studied in detail during this outbreak: 18 studies, using either Mpox-specific PCR or real-time PCR, collectively identified 606 patients that were infected solely with Mpox, a further 228 patients had dual Mpox and HIV infections, 1 patient was co-infected with Mpox, HIV, and SARS-CoV2, and a final patient was co-infected with Mpox, HIV, and syphilis [180]. An overwhelming majority of case-patients were male, with a median age of 35 years [86]. The detection of Mpox DNA in semen samples [99,138,178], along with the reported isolation of replication-competent Mpox from seminal fluid [138], suggested the possibility of sexual transmission of Mpox between humans, however, further studies are required.

Considering minor modes of human-to-human transmission, three studies [5,100,181] reported hospital-acquired transmission of Mpox, which might have occurred due to sharing of hospital beds, exposure to contaminated equipment(s) during the hospital visits, and/or spillover to the healthcare workers during inpatient treatments. One study reported mother-to-fetus transmission of Mpox [40].

Twenty-three studies reported animal-to-human zoonotic transmission of Mpox [3,4,5,6,11,61,65,66,83,85,92,109,110,112,113,114,120,126,134,141,151,153,158], which was suggested to have occurred due to several factors, including exposure to or feeding upon dead wild animals or occasionally from pets within the household. Five studies suggested animal-to-animal, and one study identified human-to-dog transmission of Mpox due to close contact or extended sharing of space [51,64,89,91,153,154] (Figure 8A). Some studies (*n* = 6) reported that close contact with pet prairie dogs might transmit Mpox to humans [5,11,44,66,91,92]. Contact with, or feeding upon, monkeys and other nonhuman primates was another common risk factor for Mpox zoonosis, reported in five separate studies [114,126,141,173,182]. Similarly, hunting and eating squirrels can transmit Mpox to humans (*n* = 3) [109,113,158]. Additionally, one study each reported that human contact with pet rodents [66], rodent carcasses [4], and dead squirrels [151] were some of the risk factors for Mpox zoonosis (Figure 8B).

Animal-to-animal transmission of Mpox may be facilitated by close contact between animals in their natural habitat [122,152] or due to housing within a restricted space [153]. In addition to the close proximity, one study [107] determined that feeding or scavenging by flies and maggots upon dead animals or carcasses may transmit Mpox (Figure 8C).

Collectively, the studies reporting zoonotic and interspecies transmission of Mpox reinforce that this is a contagious virus that combines a high potential of zoonotic spillover with a well-documented wide host range. Hosts include various species of nonhuman primates and small mammals, especially rodents. While the ‘West African’ clade of Mpox dominated the circulation within and outside the endemic regions more than the ‘Central African’, formerly known as ‘Congo Basin’, clade, the multi-country Mpox outbreaks of 2022 were suggested to have occurred due to the emergence of a new clade, termed ‘hMPXV-1A’ lineage B.1, which appeared to have diverged from the ‘West African’ clade [101]. Skin rash, fever, lesions, and lymphadenopathy were some of the most commonly occurring clinical symptoms of Mpox disease in humans; however, various other symptoms may also be present. As observed during 2022 Mpox outbreaks in non-endemic countries, clinical disease symptoms, such as proctitis and anogenital ulcers, were a few of the unusual symptoms observed in Mpox infections in humans (Appendix A). Close contact between humans and animals, as well as customary feeding upon wild animals, were observed to be major risk factors triggering Mpox zoonosis. While earlier studies detected Mpox seroprevalence using various serologic techniques, the virologic investigations were based on virus isolation, EM, and histologic examination using IHC. In later years, the introduction of DNA amplification-based detection, such as real-time PCR, became the choice of the molecular detection of Mpox in clinical specimens and successfully detected Mpox DNA in a larger number of clinical samples and organ tissues than any other methods used. Further, Sanger and the next-generation sequencing techniques, in combination with conventional PCR, greatly facilitated Mpox genome sequencing for epidemiological investigations and monitoring the evolution of the Mpox genome and the clade determination.

## 4. Discussion

The clinical symptoms of Mpox in humans may be similar to those caused by other members of the OPXV genus; thus, the molecular detection of Mpox is required to confirm the diagnosis. Interestingly, based on clinical disease symptoms and detection techniques used, human cases of Mpox can be categorized into three groups: suspected, probable, and confirmed [117]. According to the recently revised definitions of Mpox cases, suspect cases are those with a febrile prodrome with at least one or more clinical manifestations such as skin rash, headache, backache, asthenia, lymphadenopathy, myalgia, or sudden onset of anogenital complaints developed after 15 March 2022, unexplained by other differential diagnoses [117]. Mpox probable cases are those that meet the Mpox suspect case criteria and have traveled to the Mpox endemic region or had contact with an Mpox suspected, probable, or confirmed case within the past 21 days before the onset of clinical symptoms. Additionally, sexual contact with anonymous or multiple partners within 21 days before the onset of symptoms and fulfilling the definition of a Mpox suspected case classify the cases into the Mpox probable case category. Furthermore, cases where the clinical specimens have tested positive in a laboratory using Mpox- specific real-time PCR or genome sequencing are considered Mpox-confirmed cases [117].

The earliest documented Mpox outbreaks in the endemic regions (rain forests in West and Central Africa) were caused by either West African or Central African clades, based on phylogenomics and clinical manifestations of the disease [101]. While the West African clade has been reported to inflict a mild disease, the symptoms caused by the Central African clade are relatively severe [101]. The first human cases of Mpox outside endemic regions in Africa were zoonotically transmitted from Mpox-infected pet prairie dogs. Interestingly, the prairie dogs became infected upon contact with other imported species of rodents at pet shops in the Midwestern United States in 2003 that originated from Ghana as part of the exotic pet trade. While more recently, the exportation of Mpox human cases outside of Africa started in 2017, transmitted via air travelers [69], and the multi-country outbreaks of Mpox, which began in April–May 2022, appear to have been driven by sexual contact [98,99,138]. These multi-country outbreaks of Mpox have been reported to have occurred due to a newly emerged clade termed ‘hMPXV-1A’ lineage B.1 [103], which may have diverged from the West African clade of Mpox. It has been determined that certain adaptive mutations in the Mpox 2022 isolates facilitated its human-to-human transmission [103].

The incubation period of Mpox is generally between seven to fourteen days [81]; however, it has been observed to be up to 20 [48] or 21 [183] days, depending on the route of exposure [184]. Rash, fever, skin lesions, and lymphadenopathy were some of the most observed clinical symptoms of Mpox disease in humans, caused by West African and Central African clades [5,112]. Intriguingly, the 2022 Mpox outbreaks driven by the ‘hMPXV-1A’ lineage B.1, attributed to sexual contact, appear to have additional clinical symptoms which were never reported in previous infections. Some of these unusual and characteristic symptoms related to the 2022 Mpox outbreaks include proctitis (severe anal pain and/or bleeding), anorectal lesions, anal ulcers, perianal ulcers, and penile ulcers [146,174,178]. Some of these symptoms may coincide with sexually transmitted diseases in humans, such as syphilis and gonorrhoea, and thus might complicate the diagnosis [48,185]. Therefore, it is prudent to investigate and understand the epidemiology and clinical manifestations of the 2022 Mpox epidemic to mitigate further transmission.

One of the major challenges in the effective and timely diagnosis of the Mpox epidemic is the lack of an approved Mpox detection kit. One probable reason behind this would be that the human cases of Mpox were primarily restricted to endemic regions in Africa until recently and were routinely detected using OPXV-specific conventional PCR followed by Mpox-specific real-time PCR for confirmation [85]. It was only after 2017 that Mpox human cases started appearing outside Africa and were traced to travel history [70,105]. With a limited number of sporadic case reports appearing outside Africa with limited transmission, not enough attention was given to exploring the evolution and adaptation of Mpox in humans, and, therefore, Mpox surveillance outside Africa remained neglected.

Mpox can be transmitted through close contact between humans in households [3,83]. Additionally, studies have also documented the transmission through contaminated surfaces within the household [105]. Animal-to-human transmission may occur through close contact with pets or rodents [153]. Hunting and feeding upon wild animals and exposure to dead animals or carcasses are also risk factors for Mpox infections [109]. Intriguingly, maggots and flies that were observed feeding upon a dead chimpanzee in the wild were positive for Mpox DNA and suggested that scavenging may be involved in the transmission of Mpox [107]. Whether maggots and flies can serve as vectors of Mpox remains a question for further investigation.

Most cases of Mpox in animals were detected in nonhuman primates and rodents; only one study each reported Mpox DNA in a pet dog (*Canis lupus familiaris*) [154] and antibodies in a domestic pig (*Sus scrofa domesticus*) [113]. While domestic pigs are reported to harbor several virus pathogens, including RNA [186,187,188,189,190,191,192] and DNA [193,194] viruses, the occurrence of Mpox antibodies in a domestic pig indicates the possibility of natural infection of domestic pigs with Mpox and warrants the surveillance of Mpox in domestic pig populations to investigate their status and significance.

Earlier serologic studies detected OPXV antibodies in human sera, which supplemented the diagnosis in the presence of the typical poxvirus disease symptoms, such as the presence of maculopustular rash, skin lesions, fever, lymphadenopathy, etc. While IgG ELISA was one of the most used serologic methods to detect past exposure to Mpox in humans, IgM ELISA detected more recent infections in humans. Most studies that used ELISA detected OPXV antibodies and, based on clinical symptoms, suspected previous exposures to Mpox [123,195]. Most of these earlier studies, solely based on serologic detection, could only suggest the occurrence of Mpox probable cases but, in most cases, were not able to confirm the presence of Mpox because some of the other poxviruses, such as smallpox, vaccinia, and variola, are known to cause similar clinical symptoms in humans [123]. Thus, more sensitive and reliable detection methods were required to make a conclusive diagnosis. In the absence of DNA-based detection, several of those earlier studies used EM [11,12,13] and virus isolation [11,12] methods to confirm the Mpox disease diagnosis. While virus isolation is a gold standard for virological investigations [37], success depends on multiple factors, including virus titer in the sample and sample handling, such as collecting and storing samples.

Similarly, EM is a traditional gold standard for the morphological identification of emerging and novel viruses [37] and has been proven quite useful for distinguishing isolated Mpox particles from other OPXV particles [60,61]. In addition, histological methods such as IHC were used in some studies to detect Mpox antigen in stained tissues obtained from an autopsy. IHC is a useful technique for understanding the distribution of virus particles in infected tissues [196].

The advent of PCR techniques in later years greatly facilitated the targeted detection of Mpox DNA in various clinical and environmental samples. While real-time PCR detected the highest number of Mpox infections in human populations, some of the molecular studies used genome sequencing utilizing Sanger or next-generation sequencing techniques, which greatly facilitated the monitoring of the Mpox genome evolution [4,9]. One of the greatest challenges of detecting Mpox in suspected human cases was the lack of approved or recommended Mpox detection kits by authorities such as the WHO or CDC. The scale and magnitude of the 2022 Mpox outbreaks require the availability of an automated Mpox testing system, such as the GeneXpert MPX/OPX platform, which is reported to be under development [128]. This system can detect Mpox using an OPXV-generic and Mpox-specific multiplex detection approach with high sensitivity and 100% specificity and has been tested under laboratory and field conditions in the initial phase [128]. The availability of a high-throughput and scalable, automated system for Mpox detection in an epidemic setting would greatly reduce the turnaround time of testing with the added advantage of minimal manual sample handling to reduce the risk of exposure to laboratory personnel.

In the current scenario, real-time PCR-based detection of Mpox appears cost-effective and accessible for routine diagnostic testing. Real-time PCR is the most common method used for Mpox detection and is also considered a gold standard in diagnostic laboratories. While various real-time PCR assays may vary in the gene targets, specificity, and sensitivity, they still identified a much higher number of Mpox-positive human cases than any other methods used. The advantage of molecular testing of Mpox over serologic testing is that varied sample types can be used for PCR-based detection of Mpox. For example, a large number of studies successfully detected Mpox in skin lesion swabs, crusts, and blood [6,86,88,116,170], but on the other hand, serologic investigations could only use blood for the detection of OPXV antibodies [124,133,195]. A rapid, reliable, and cost-effective molecular assay for Mpox detection would provide more sensitive and specific detection over serologic methods. In addition, genomic surveillance of Mpox using genome sequencing will generate Mpox genomes which will help update the molecular diagnostic tools and existing vaccines to safeguard human and veterinary health. Development of a rapid, sensitive, OPXV and Mpox-specific multiplex real-time PCR is prudent for scalable detection of Mpox while detecting and differentiating among poxviruses in clinical samples. The increasing use of next-generation sequencing methods in recent years, such as Illumina and Oxford Nanpore MinION, appeared promising regarding genomic surveillance of Mpox, generated complete Mpox genomes using clinical samples, and therefore, were instrumental in determining the emergence of a new Mpox clade, hMPXV-1A lineage B.1, during 2022 Mpox outbreaks in non-endemic countries.

## 5. Conclusions

Real-time PCR offers rapid and reliable detection of Mpox in clinical samples; however, there is scope for developing a multiplex real-time PCR assay for simultaneous detection of commonly circulating poxviruses and for OPXV-generic and Mpox-specific detection in a single assay. It would be interesting to see if sequencing platform(s) could offer cost-effective detection of poxviruses based on genome sequences in the coming years. Mpox genome sequencing would be informative and useful for Mpox epidemic preparedness, for updating diagnostic methods such as oligonucleotide primers, probes, and antibodies, for superior sensitivity and specificity of assays in the detection of (re)emerging Mpox strains or variants. Mpox genome sequencing will enable more efficient and real-time tracking of the novel variants and emerging strains for accurate epidemiological analysis and will assist in identifying vaccine candidates. Though not recommended by the WHO for routine diagnosis of Mpox in clinical laboratories, virus isolation remains particularly useful for detecting and investigating the viability, pathogenicity, and virulence of Mpox in clinical samples. Moreover, the EM remains useful for the differentiation and morphological characterization of poxviruses. The IHC remains another useful method for investigating Mpox systemic infection and antigen localization in various animal tissues and organs. The IgG and IgM ELISA remain useful for serologic detection investigating recent or past Mpox infections while determining vaccine efficiencies in a given population, especially in Mpox endemic regions.

## Figures and Tables

**Figure 1 microorganisms-11-01186-f001:**
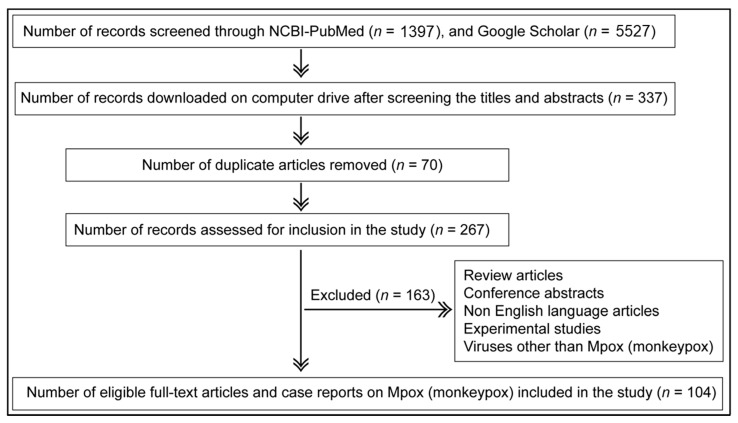
PRISMA chart depicting the search criteria for identifying original research articles and case reports reporting natural infections of humans and/or animals with Mpox for inclusion in the study. A total of 104 eligible full-text research articles and case reports were included in the study.

**Figure 2 microorganisms-11-01186-f002:**
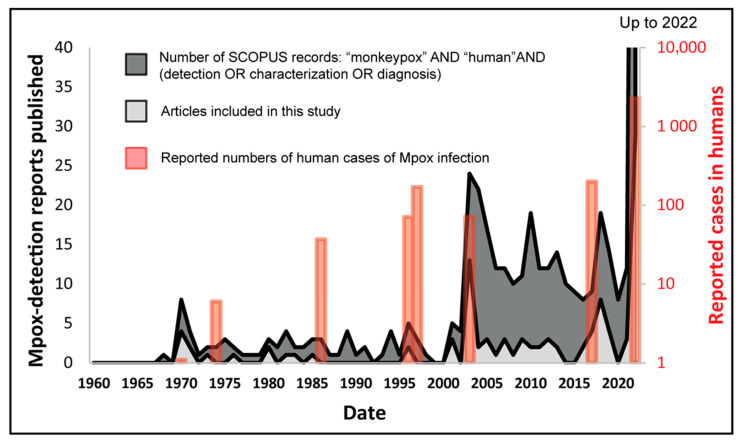
Progression of Mpox disease surveillance in humans. The highest number of studies detecting Mpox infections were conducted during multi-country outbreaks in 2022. Estimates of cases were drawn from selected published reports, including the following citations [48,58,83,84,85,86,87,88].

**Figure 3 microorganisms-11-01186-f003:**
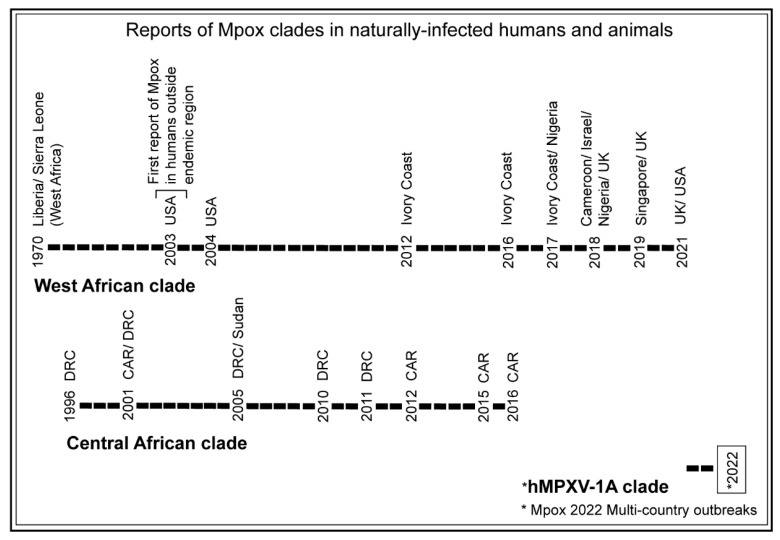
Historically, a higher circulation frequency of the West African clade of Mpox occurred [4,7,10,11,42,65,68,70,71,83,90,104,105,106,107,108,109,110] than the Central African clade [6,9,13,111,112,113,114,115]. The 2022 Mpox outbreaks were caused by a newly emerged clade termed ‘hMPXV-1A’ lineage B.1 [103] and suggested the ongoing evolution of Mpox (CAR = Central African Republic; DRC = Democratic Republic of the Congo; UK = United Kingdom; USA = United States of America). *Mpox 2022 multi-country outbreaks were caused by a newly emerged clade ‘hMPXV-1A’ lineage B.1.

**Figure 4 microorganisms-11-01186-f004:**
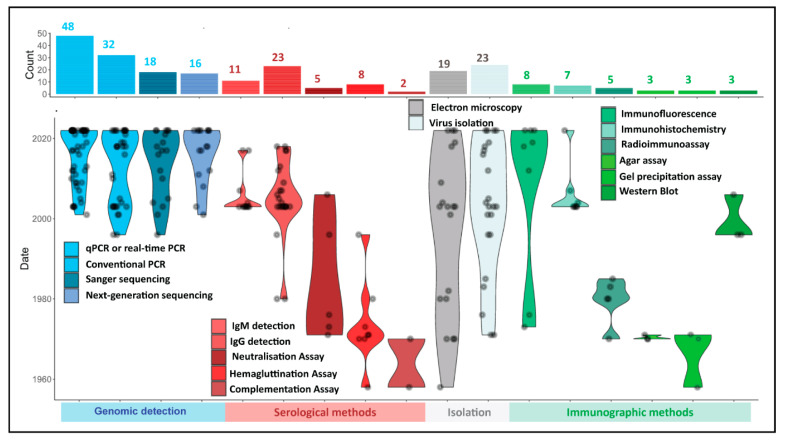
A schematic representation of various methods used for Mpox detection in humans and animals over the period (1958–2022).

**Figure 5 microorganisms-11-01186-f005:**
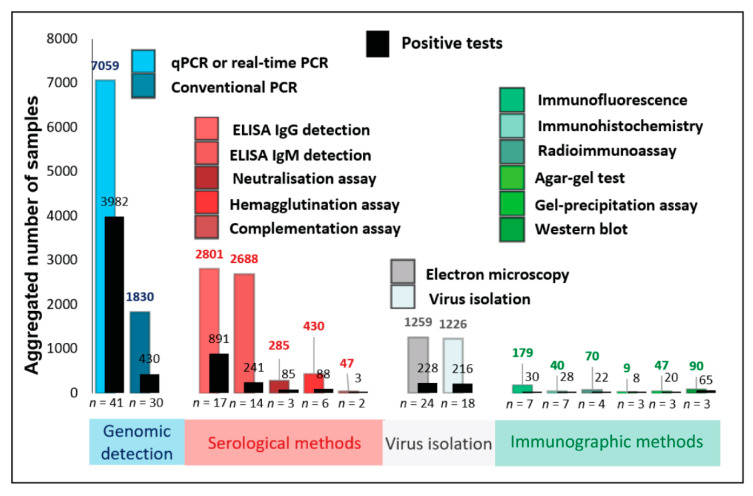
Representation of all the cases investigated for Mpox or OPXV detection using various methods in human clinical samples from eligible articles available up to 2 September 2022; *n* = number of studies.

**Figure 6 microorganisms-11-01186-f006:**
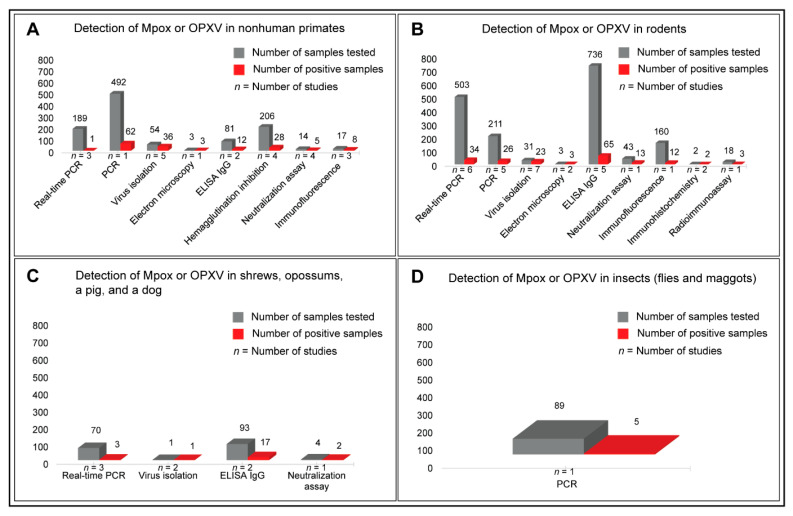
Detection of Mpox in animals. (**A**) More Mpox infections in nonhuman primates were detected using conventional PCR than serologic methods and virus isolation. (**B**) Mpox infections were successfully detected in various species of rodents using molecular and serologic methods, virus isolation, IHC, and EM. (**C**) Mpox was also detected in some other mammals, including shrews, opossums, a pig, and a dog using real-time PCR, serologic assays, and virus isolation. (**D**) Conventional PCR detected Mpox infection in some of the flies and maggots feeding upon a chimpanzee with external Mpox lesions found dead in the forest.

**Figure 7 microorganisms-11-01186-f007:**
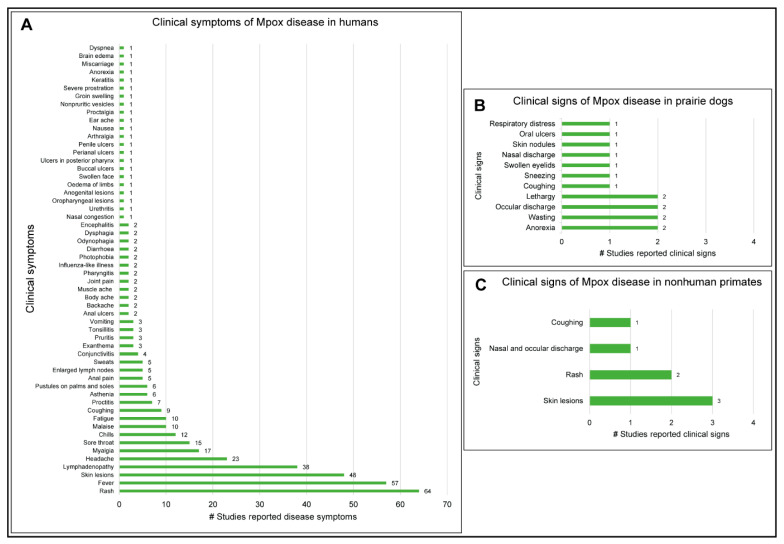
Number of studies that reported (**A**) the clinical symptoms of Mpox disease in humans, (**B**) clinical signs of Mpox disease in prairie dogs, and (**C**) clinical signs of Mpox disease in nonhuman primates.

**Figure 8 microorganisms-11-01186-f008:**
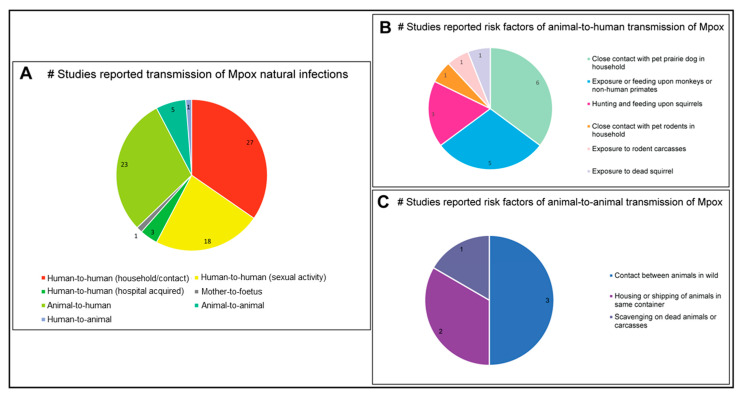
Transmission dynamics of Mpox. (**A**) Zoonotic or reverse zoonotic transmission of Mpox in humans and animals. (**B**) Risk factors involved in zoonotic transmission of Mpox. (**C**) Risk factors involved in the animal-to-animal transmission of Mpox.

**Table 1 microorganisms-11-01186-t001:** Search terms used for identifying the relevant research articles and case reports of Mpox natural infections in humans and animals up to 2 September 2022.

Search Terms Used	# Titles Screened in NCBI-PubMed	# Titles Screened in Google Scholar
Monkeypox virus outbreak	328	430
Detection of monkeypox virus disease	83	390
Monkeypox virus in Africa	240	330
Epidemiology of monkeypox virus disease	223	270
Molecular detection of monkeypox virus	33	310
Serological detection of monkeypox virus	10	270
Microarray for monkeypox virus	15	210
Real-time PCR for monkeypox virus	43	360
Antigenic detection of monkeypox virus	29	250
Genomic sequencing of monkeypox virus	83	220
Rapid detection of monkeypox virus	28	260
Point of care detection of monkeypox virus	3	150
ELISA for monkeypox virus detection	16	190
Monkeypox virus outbreak 2022	263	490

**Table 2 microorganisms-11-01186-t002:** Oligonucleotide primers and probes used for Mpox detection in clinical samples.

Target (Specificity)	ORF or Genomic Region/Primer and Probe ID	Oligonucleotide Sequence 5′–3′	References (Earliest Citation)
General OPXV-detecting real-time PCR			
P4A: Major core protein 4a precursor (OPXV family)	P4A/Forward	TAATACTTCGATTGCTCATCCAGG	[155]
	P4A/Reverse	ACTTCTCACAAATGGATTTGAAAATC	
	P4A/Probe	FAM-TCCTTTACGTGATAAATCAT-NFQ MGB	
E9L: DNA polymerase gene (OPXV family; non-variola)	E9L/Forward	TCAACTGAAAAGGCCATCTATGA	[13,64,114,153,156]
	E9L/Reverse	GAGTATAGAGCACTATTTCTAAATCCCA	
	E9L/NVAR-Probe	TET-CCATGCAATATACGTACAAGATA- GTAGCCAAC-QSY7	
Mpox-specific real-time PCR			
G2R: TNF receptor gene (Mpox-generic)	G2R/G-Forward	GGAAAATGTAAAGACAACGAATACAG	[3,8,114,157,158]
	G2R/G-Reverse	GCTATCACATAATCTGGAAGCGTA	
	G2R/G-Probe	FAM-AAGCCGTAATCTATGTTGTCTATC-GTGTCC-BHQ1	
G2R: TNF receptor gene (West African-specific Mpox clade)	G2R/WA-Forward	CACACCGTCTCTTCCACAGA	[3,8,114,157,158]
	G2R/WA-Reverse	GATACAGGTTAATTTCCACATCG	
	G2R/WA-Probe	FAM-AACCCGTCGTAACCAG-CAATACATTT-BHQ1	
C3L: Complement binding protein Central African (formerly Congo Basin)-specific Mpox clade	C3L/CB-Forward	TGTCTACCTGGATACAGAAAGCAA	[3,8,114,157,158]
	C3L/CB-Reverse	GGCATCTCCGTTTAATACATTGAT	
	C3L/CB-Probe	FAM-CCCATATATGCTAAATGTACCGGT-ACCGGA-BHQ1	
B6R: Envelope protein gene (Mpox)	B6R/Forward	ATTGGTCATTATTTTTGTCACAGGAACA	[13,64,114,153,156]
	B6R/Reverse	AATGGCGTTGACAATTATGGGTG	
	B6R/Probe	MGB-AGAGATTAGAAATA-FAM	
F3L: Interferon resistance gene (Mpox)	F3L/F290-Forward	CTCATTGATTTTTCGCGGGATA	[40,89,109,159]
	F3L/R396-Reverse	GACGATACTCCTCCTCGTTGGT	
	F3L/p333S-MGB-Probe	FAM-CATCAGAATCTGTAGGCCGT-MGB NFQ	
N3R: Conserved (unclassified function) (Mpox)	N3R/F319-Forward	AACAACCGTCCTACAATTAAACAACA	[40,89,109,159]
	N3R/R457-Reverse	CGCTATCGAACCATTTTTGTAGTCT	
	N3R/p352S-MGB-Probe	FAM-TATAACGGCGAAGAATATACT-MGB NFQ	
Conventional Mpox PCR			
E5R: Nucleic acid-independent nucleoside triphosphatase (Mpox)	E5R/Forward	ATGTTGATATTAATAATCGTATTGTGGTT	[4,160]
	E5R/Reverse	AAAGTCAATACACTCTTAAAGATTCTCAA	
ATI: A-type inclusion body protein (Mpox)	ATI/Gabon 1- Forward	GAGAGAATCTCTTGATAT	[142,161]
	ATI/Gabon 2- Reverse	ATTCTAGATTGTAATC	

Primer-conjugated fluorophores and quenchers appear at 5’ and 3’ ends, respectively, in the above sequences, with the exception of B6R (envelope protein gene) of Mpox. Primer-bound fluorophores: FAM—6-carboxyfluoresceine; TET—tetrachlorofluoresceine. Primer-bound fluorophore quenchers: MGB—Minor groove binding; QSY7—quencher succinimidyl ester; BHQ1—blackhole quencher 1.

## Data Availability

Not applicable.

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
