# Peer review of "Overview of Diagnostic Methods, Disease Prevalence and Transmission of Mpox (Formerly Monkeypox) in Humans and Animal Reservoirs"

_microorganisms, 2023, doi:10.3390/microorganisms11051186_

Round 1
Reviewer 1 Report
Dear Authors,
During reviewing your interesting paper, I found some points that need to be addressed:
1. In the Title: (monkeypox) changed to (formerly monkeypox)
2. In the Keywords: monkeypox virus changed to Mpox virus
3. L52: Of the OPXV members, Mpox is the most pathogenic virus … Is it correct?
4. L61: The Mpox disease was first identified in 1958 in cynomolgus monkeys … Mpox virus may be older than this date. Please, check these citations for this part (https://doi.org/10.1111/tbed.14651 - https://doi.org/10.3390/vaccines10122091)
5. L68-69: Due to its first detection in monkeys, term ‘monkeypox virus’ was coined in 1958, which in November 2022 was redesignated “Mpox” by the WHO [1] …. Inappropriate reference
6. L100: due to the adaptation of Mpox to human hosts … Please add a suitable reference
7. L242: 246: In general, molecular detection methods have several advantages over serological methods. Primarily, molecular detections are routinely capable of differentiating between Mpox and other species of OPXV, unlike most serological assays used [121,122]. Although a few studies used Mpox-specific antibodies, they were reported to lack reproducibility and are complex to execute [123]. …. Please updated references here for human cases.
8. I didn’t see any references to the first paragraph sentences in section: 3.8. Transmission dynamics of Mpox disease.
9. L631: Mpox is a highly contagious disease … Is this statement correct?
10. Please, remove repeated sentences like:
L625: Interestingly, one study determined that feeding or scavenging by flies and maggots upon dead animals or carcasses may transmit Mpox.
L633: Intriguingly, Mpox DNA was also detected in maggots and flies feeding/scavenging upon a dead chimpanzee in the forest, in one study.
11. The redundancy weakens the power of review.
L690: The incubation period of Mpox is between seven to fourteen days [75]. .. it can reach t21 days see (https://doi.org/10.3390/vaccines10122091).
12. There are minor typo errors.
Review should include the preferred samples for mpox diagnosis.
1. There are minor typo errors.
Author Response
Please find the attached letter.
Thank you!

Reviewer 2 Report
In this review, the authors have comprehensively described the technological advancements in Mpox detection in naturally infected human and animal populations, along with clinical manifestations and transmission dynamics of disease.
The review is well written and appears complete.
Only two minor points:
-) With reference to the 2022 outbreak, the authors should better describe the prevalence and epidemiological aspects (for example: mode of transmission, age, sex, co-infections such as HIV infection or others STIs) by adding the latest updated WHO data.
-) I ask the authors what can be the significance of the cycle threshold (CT ) values ​​ since some recent studies have reported that low values ​​of the Ct-values ​​are associated with greater viraemia and more severe clinical manifestations.
Good
Author Response

(The authors gave the same response as above.)
